# Connexin 43 Channels in Osteocytes Are Necessary for Bone Mass and Skeletal Muscle Function in Aged Male Mice

**DOI:** 10.3390/ijms232113506

**Published:** 2022-11-04

**Authors:** Guobin Li, Lan Zhang, Zhe Lu, Baoqiang Yang, Hui Yang, Peng Shang, Jean X. Jiang, Dong’en Wang, Huiyun Xu

**Affiliations:** 1Key Laboratory for Space Bioscience and Biotechnology, School of Life Sciences, Northwestern Polytechnical University, Xi’an 710072, China; 2College of Life Sciences, Inner Mongolia Agricultural University, Hohhot 010018, China; 3Key Laboratory for Space Bioscience and Biotechnology, Research and Development Institute in Shenzhen, Northwestern Polytechnical University, Shenzhen 518057, China; 4Department of Biochemistry and Structural Biology, University of Texas Health Science Center, San Antonio, TX 78229, USA

**Keywords:** osteocytes, hemichannels, gap junctions, bone-muscle crosstalk, aging

## Abstract

Osteoporosis and sarcopenia (termed “Osteosarcopenia”), the twin-aging diseases, are major contributors to reduced bone mass and muscle weakness in the elderly population. Connexin 43 (Cx43) in osteocytes has been previously reported to play vital roles in bone homeostasis and muscle function in mature mice. The Cx43-formed gap junctions (GJs) and hemichannels (HCs) in osteocytes are important portals for the exchange of small molecules in cell-to-cell and cell-to-extracellular matrix, respectively. However, the roles of Cx43-based GJs and HCs in both bone and muscle aging are still unclear. Here, we used two transgenic mouse models with overexpression of the dominant negative Cx43 mutants primarily in osteocytes driven by the 10-kb Dmp1 promoter, R76W mice (inhibited gap junctions but enhanced hemichannels) and Δ130–136 mice (both gap junction and hemichannels are inhibited), to determine the actions of Cx43-based hemichannels (HCs) and gap junctions (GJs) in the regulation of bone and skeletal muscle from aged mice (18 months) as compared with those from adult mice (10 months). We demonstrated that enhancement of Cx43 HCs reduces bone mass due to increased osteoclast surfaces while the impairment of Cx43 HCs increases osteocyte apoptosis in aged mice caused by reduced PGE_2_ levels. Furthermore, altered mitochondrial homeostasis with reduced expression of Sirt-1, OPA-1, and Drp-1 resulted in excessive ROS level in muscle soleus (SL) of aged transgenic mice. In vitro, the impairment of Cx43 HCs in osteocytes from aged mice also promoted muscle collagen synthesis through activation of TGFβ/smad2/3 signaling because of reduced PGE_2_ levels in the PO CM. These findings indicate that the enhancement of Cx43 HCs while GJs are inhibited reduces bone mass, and the impairment of Cx43 HCs inhibits PGE_2_ level in osteocytes and this reduction promotes muscle collagen synthesis in skeletal muscle through activation of TGFβ/smad2/3 signaling, which together with increased ROS level contributes to reduced muscle force in aged mice.

## 1. Introduction

Bone and skeletal muscle are the two important integrated components within the musculoskeletal system. Osteoporosis and sarcopenia featured by the decline of bone mass and muscle function, respectively, are inevitable consequences of an aging musculoskeletal system, which further increase the risks of fracture, falls, and low quality of life in the elderly [1,2]. As the world’s population ages, the incidence of musculoskeletal diseases is increasing and it has raised a significant public health concern; therefore, it is necessary to understand these two aging-related conditions to prevent both bone and muscle weakness.

Previous studies have demonstrated that an intimate developmental and functional linkage exists between bone and skeletal muscle [3,4]. In particular, there is increasing support for the concept that bone and muscle act as secretory organs producing “hormone-like factors” that can mutually affect each other [5,6] and other distant organs [7,8]. Evidence has emerged that many bone-specific cytokines have profound effects on muscle cells in vitro [1,4,9,10,11], and vice versa [2,12,13,14], indicating that the bidirectional biochemical communication between bone and muscle is important for their optimal function.

Osteocytes, embedded within the bone matrix, make up 90–95% of total bone cells in the adult skeleton. Their long dendritic processes form an extensive network that allows for the communication between neighboring osteocytes and with other cells on the bone surface [15]. Gap junctions (GJs) and hemichannels (HCs) both formed by connexin 43 (Cx43) in bone mediate the communication of cell-to-cell and cell-to-extracellular space, respectively. These connexin-based channels are only permeable to small molecules (MW < 1.2 kDa), including adenosine triphosphate (ATP), cyclic adenosine monophosphate (cAMP), nitric oxide (NO), prostaglandin E2 (PGE2) and some metabolites [16], which also increases the possibility that osteocytes interact with skeletal muscle through the release of osteokines to the lacunar–canalicular system. PGE2, derived from arachidonic acid metabolism by cyclooxygenase [17], has been proved to inhibit collagen synthesis in tissue fibrosis. The expression of transforming growth factor-β (TGF-β) was positively correlated with collagen synthesis through the activation of the downstream Smad2/3 protein. However, the roles of Cx43-based channels in its regulation through PGE2 in muscle collagen synthesis are still unclear.

Many studies have illustrated the importance of Cx43 in early bone development [18] and mature bone remodeling [19,20]. Plotkin and colleagues found Cx43 deficiency in osteocytes partially mimics the skeletal phenotypes of old mice with enhanced endocortical resorption and increased osteocyte apoptosis [21]. Furthermore, another study by Plotkin’s lab has revealed that increased Cx43 level in osteocytes prevents cell death and maintains bone quality in aging mice [22]. However, their mouse models fail to dissect the specific involvement of Cx43-formed GJs and HCs in age-related bone phenotypes. To this end, we used two transgenic mouse models driven by 10-kb fragment of the dentin matrix protein 1 (Dmp1) promoter with the overexpression of Cx43 mainly in osteocytes: R76W, which harbors dominant-negatively blocked GJs albeit enhanced HCs, and Δ130–136 has dominant-negative effects on both GJs and HCs. Using the two transgenic mice, our previous studies have well-distinguished the differential functions of GJs and HCs formed by Cx43 in bone remodeling [23], fracture healing [24], bone loss induced by estrogen deficiency [16], and bone response to mechanical unloading [25].

In this study, we used 10- and 18-month-old transgenic mouse models expressing dominant-negative Cx43 mutants, R76W and Δ130–136, to examine the functional contribution of Cx43-formed GJs and HCs in osteocytes to the phenotypes of bone and skeletal muscle in adult and aged mice. Herein, we showed that impairment of Cx43 HCs in osteocytes displays a protective effect on bone mass but compromises skeletal muscle function due to increased muscle collagen deposition in aged mice. Our findings have implications for bone-muscle crosstalk during aging and potential therapeutic approaches for treatment of sarco-osteoporosis and other consequences of aging.

## 2. Results

### 2.1. Enhancement of HCs while GJs Are Inhibited Reduced Bone Mass in Older Mice

Three-dimensional μCT was used to determine the effect of osteocytic Cx43 channels on the skeletal microstructure with age. Representative μCT images of femoral trabecular bone in 10- and 18-month-old mice are illustrated in Figure 1a. The trabecular parameters of the distal femur, including BV/TV (Figure 1b) and Tb.N (Figure 1c), were significantly decreased associated with a significant increase in Tb.Sp (Figure 1d) in R76W compared to Δ130–136 mice at 18 months of age, without changes in Tb.Th (Figure 1e). However, all these trabecular parameters remained unaltered in all three genotypes in 10-month-old mice (Figure 1a–e). Further, BV/TV (Figure 1b) and Tb.N (Figure 1c) in R76W and WT mice at age of 10 months were significantly higher than that of 18 months, with a significant decrease in Tb.Sp (Figure 1d). Similarly, representative μCT images of cortical bone in 10- and 18-month-old mice are illustrated in Figure 1f. The μCT analysis showed a significant decrease in Ct.Ar (Figure 1g) and Ct.Th (Figure 1h) in R76W compared to that of WT and Δ130–136 mice at 18 months of age, without changes in Ma.Ar (Figure 1i) and B.Ar/T.Ar (Figure 1j). Moreover, these cortical bone parameters also remained unaltered across groups in 10-month-old mice. Further, 18-month-old transgenic mice exhibited a significant decrease of Ct.Th (Figure 1h) and B.Ar/T.Ar (Figure 1j) compared to 10-month-old transgenic mice, respectively. Collectively, given the functional differences between R76W and Δ130–136 in the two channels, these results indicate that the enhancement of HCs while GJs are inhibited resulted in reduced bone mass in older mice.

### 2.2. Increased Cortical Osteocyte Apoptosis and Empty Lacunae in Δ130–136 Mice

Analysis of H&E stained sections showed an increase in the proportion of empty lacunae in cortical bone in 10- and 18-month-old Δ130–136 mice, but this increase was not observed in age-matched R76W mice (Figure 2a,c). Additionally, there were no alterations of N.Ot/B.Ar in all three genotypes at age of 10 and 18 months (Figure 2b). Consistently, immunohistochemistry staining was performed to further analyze the number of apoptotic osteocytes expressing active caspase-3 (Figure 2d) and showed that Δ130–136 mice exhibited an increase of active caspase-3-positive osteocytes compared to WT and R76W mice (Figure 2e). Overall, these results suggest that osteocytes of cortical bone in Δ130–136 were more vulnerable to age-related apoptosis than those in R76W and WT mice, indicating a protective effect of Cx43 HCs on osteocyte viability.

### 2.3. Cx43 Channels Affect Activity of Osteoclasts and Osteoblasts in Older Mice

The TRAP staining showed that endocortical osteoclast number and osteoclast surfaces were both significantly increased in 18-month-old R76W mice compared to that of Δ130–136 mice (Figure 3a–c). Consistent with these observations, the mRNA expression of NFATc-1 (Figure 3d) was significantly increased in R76W compared to Δ130–136 mice at age of 18 months. Data also showed increased circulating ATP in 18-month-old R76W compared to that of Δ130–136 mice (Figure 3e). In addition, H&E staining showed osteoblast number was significantly reduced in 18-month-old Δ130–136 compared to that of R76W and WT mice (Figure 3f). At the mRNA level, the expression of Alp was also markedly reduced in 18-month-old Δ130–136 mice (Figure 3g). However, these observations were not observed across groups at age of 10 months. Therefore, these data suggest that enhancement of HCs while GJs are inhibited increases osteoclast surfaces and osteoclast number and impairment of HCs reduced osteoblast number in older mice.

### 2.4. Muscle Mass and Myofiber Composition Was Not Altered in Older Transgenic Mice

To determine whether functional alterations of osteocytic Cx43 channels also affect myofiber morphology and muscle function with age, we further measured muscle mass, myofiber size, and muscle function in 10- and 18-month-old transgenic mice. As illustrated in Appendix A, SL muscle mass was significantly increased only in 10-month-old R76W mice, which was explained, at least in part, by an increase in myofiber cross-sectional area (CSA) at age of 10 months (Appendix A). However, SL muscle mass and myofiber CSA did not differ between transgenic mice and their WT littermates at age of 18 months (Appendix A–c). In addition, there was also no significant difference in SL myofiber number across groups at age of 10 or 18 months (Appendix A). We next performed myosin heavy chain typeI (MHCI) staining on sections of SL in 10- and 18-month-old transgenic mice (Appendix A) and found that the proportion of typeI fibers (Appendix A) and myofiber I CSA (Appendix A) also did not differ in all three genotypes. Correspondingly, the mRNA level of Myh7, the gene coding MHC I, was also not significant across groups in 10- and 18-month-old mice, respectively (Appendix A).

### 2.5. Impairment of Cx43 HCs Decreases Muscle Contractile Force in Older Mice

Ex vivo SL muscle contractile force was performed, and we observed that the maximal absolute isometric twitch tension (Figure 4a), maximal specific isometric twitch tension (Figure 4b), maximal absolute tetanic tension (Figure 4c), and maximal specific tetanic force (Figure 4d) were significantly reduced in Δ130–136 compared to R76W and WT mice at age of 18 months. However, these observations were not observed in 10-month-old transgenic mice. Therefore, the impairment of Cx43 HCs results in decreased muscle function in older mice.

### 2.6. Effects of Osteocytic Cx43 Channels on Oxidative Stress and Mitochondrial Homoeostasis in Muscle Aging

To further explore the potential reasons for the impaired skeletal muscle function in 18-month-old transgenic mice, we also evaluated the effects of Cx43 channels on oxidative stress parameters in SL muscle of aged transgenic mice. The ROS level was markedly elevated both in R76W and Δ130–136 mice compared to WT (Figure 5a). Next, we investigated the changes in levels of enzymatic antioxidants in aged muscle; we observed that SOD activity was significantly decreased in the transgenic mice (Figure 5b). Additionally, catalase (CAT) activity (Figure 5c) and glutathione (GSH) level (Figure 5d) were both significantly decreased in Δ130–136 mice compared to WT mice. Mitochondria are an important source of excess ROS in aging muscle. We further determined the changes in the expression of mitochondrial proteins, and data showed that the expression of protein associated with fusion (optic atrophy type 1; OPA-1) (Figure 5e,f) and mitochondrial biogenesis (Sirtuin 1; Sirt-1) (Figure 5e,h) of SL muscles were significantly decreased in aged Δ130–136 mice compared to R76W and WT, respectively. Similarly, the protein expression of fission (dynamin-related protein 1; Drp-1) was also decreased in aged Δ130–136 mice (Figure 5e,g). Collectively, impairment of osteocytic Cx43 channels may affect mitochondrial homoeostasis, resulting in increased ROS level and decreased antioxidant enzyme activity in muscle aging.

### 2.7. Impairment of Osteocytic Cx43 HCs Promotes Fibrotic Phenotype in Muscle Aging

Sirius red staining showed that evident collagen deposition in the SL muscle, especially in the space between muscle cells, was observed in 18-month-old Δ130–136 mice (Figure 6a), and the collagen area fraction was significantly higher than that in the R76W and WT mice (Figure 6b). Western blotting was further performed to analyze the expression of collagenI (Col I) and TGFβ1, the most abundant components in the intramuscular connective tissue [26]. Consistent with the observation of Sirius red staining, the protein expression levels of Col I (Figure 6c,d) and TGFβ1 (Figure 6c,e) in SL muscles were significantly increased in aged Δ130–136 mice. These findings suggest that impairment of HCs increased the expression of Col I and TGFβ1 in the aged muscles, which largely leads to a fibrotic phenotype.

### 2.8. PGE_2_ Inhibits Collagen Synthesis in C2C12 Cells Treated by Aged Δ130–136-PO CM

We further detected PGE_2_ levels in the primary osteocyte conditioned media (PO CM) from aged transgenic mice and their WT littermates. In vitro results showed that PGE_2_ levels were reduced in Δ130–136-PO CM compared to R76W and WT (Figure 6f). Based on these findings and previous observations, we speculated that PGE_2_ is possibly responsible for collagen synthesis in aged muscle of Δ130–136 mice. In vitro experiments were further performed to test the protein expression of collagen synthesis in C2C12 myoblast treated by Δ130–136-PO CM or Δ130–136-PO CM supplemented with PGE_2_. As shown in Figure 6g–i, myoblast cultures treated with Δ130–136-PO CM showed an increase in the expression of Col I and TGFβ1 compared to that of WT-PO CM. Interestingly, this increase was significantly restrained after supplementing PGE_2_ in the Δ130–136-PO CM. Consistently, treatment with aged Δ130–136-PO CM also increased the phosphorylation level of smad2/3, the downstream signaling molecules of TGFβ, however, PGE_2_ supplement in Δ130–136-PO CM inhibited this increase (Figure 6j). Therefore, these findings indicated that PGE_2_ inhibits collagen synthesis in C2C12 myoblasts treated with aged Δ130–136-PO CM.

## 3. Discussion

Previous studies by Plotkin’s group have illustrated that Cx43 is critical for osteocyte viability and overexpression of Cx43 in osteocytes can ameliorate age-related cortical bone changes by maintaining osteocyte survival and bone quality [22]. These findings lead us to speculate that the Cx43 channels in osteocytes may also have an important influence on bone aging. The μCT analysis showed here that bone mass was only reduced in 18-month-old R76W mice compared with age-matched Δ130–136 mice. The data were in stark contrast to our previous work in which no differences were observed in bone mass across all three genotypes in 4-month-old mice [23]. This in vivo evidence implies the distinctive roles of osteocytic HCs or GJs in mice at different ages. The Cx43-formed HCs in osteocytes are essential for communication between osteocytes and osteoclasts [27]. There is evidence that the opening of HCs promotes the release of ATP and other small molecules from osteocytes [28,29]. We think enhancement of Cx43 HCs in R76W mice may affect osteoclast surfaces and subsequent reduced bone mass via the release of ATP, a regulatory factor for osteoclast formation and bone resorption [30,31,32]. Bone formation is also important for bone remodeling of bone aging [33]. In this study, a significant reduction was observed in osteoblast number in old Δ130–136 mice with normal bone mass, which may be attributed to that impairment of Cx43 HCs counteracts the delay of osteoblastogenesis regulated by osteoblasts. However, this observation remains to be further confirmed by the measurements of bone formation rate.

We showed augmentation of osteocyte death as evidenced by empty lacunae and increased apoptotic signals in old Δ130–136 mice. The involvement of Cx43 has been elucidated; osteocyte-specific deficient Cx43 mice showed increased osteocyte apoptosis in cortical bone, resulting in impaired bone strength [21,34]. Notably, findings from Plotkin et al. [35] revealed that the anti-apoptotic effects of bisphosphonates, drugs for treatment of osteoporosis, on osteocytes are mainly mediated via activation of Cx43 HCs. Many in vitro studies also have demonstrated that Cx43 HCs can protect osteocytes from cell death and inhibition of HCs with anti-Cx43 E2 antibody in osteocytes accelerates H_2_O_2_-induced apoptosis [16,36], suggestive of a protective role of Cx43 HCs for osteocyte survival. Consistently, our study showed that impairment of HCs led to an increase in the empty lacunae and expression level of active caspase-3, a central executor of the apoptotic pathway [37]. Moreover, previous studies by our group and others have proved that PGE2 released through Cx43 HCs could protect osteocytes against glucocorticoid-induced cell death [28,38]. Consequently, we considered the increase of empty lacunae and apoptotic osteocytes in old Δ130–136 mice could be explained by the reduced PGE2 release due to the impairment of HCs. In addition, Cx43 also performs channel-independent function to maintain bone homeostasis [39], suggesting a complex regulatory mechanism between osteocyte survival and bone mass. This may partially account for the variable results.

Osteocytes, as endocrine cells, also play an important role in bone-muscle crosstalk [1,5,40]. The findings in our recent study indicate that impairment of Cx43-based HCs and GJs in osteocytes respectively decreases fast-twitch muscle mass and muscle contractile force in young mice [41]. In this study, however, except for the decrease of slow-twitch muscle contractile force in old Δ130–136 mice, we did not observe additional changes in muscle mass or myofiber size. Previous studies have demonstrated that muscle mass is not necessarily directly correlated with muscle function [42]. Studies have indicated that aging increases intramuscular fibrous tissues composed of collagen, which further enhances muscle stiffness and impairs muscle function [43,44]. Collagen synthesis in aging muscle mainly depends on growth factors including TGFβ, interleukin (IL)-6, IL-8, fibroblast growth factors (FGFs), and vascular endothelial growth factor (VEGF). Among these factors, TGFβ1 is known to induce the collagen synthesis through activation of downstream smad2/3 signaling [43]. In this study, we also observed apparent collagen deposition in aging muscle of Δ130–136 mice, which may be related to the high expression of Col I and TGFβ1. Therefore, osteocytic HCs appear to play a protective role against the decline in muscle function caused by increased muscle collagen accumulation during aging.

The anti-fibrotic lipid mediator PGE2 is mainly derived from arachidonic acid (AA) through the cyclooxygenase pathway [45]. It was previously reported that low PGE2 level was observed in human pulmonary fibrosis [46] and animal models of fibrosis [47]. The reduction of PGE2 inhibits collagen synthesis and is mainly mediated by the activation of its receptors, EP2 and EP4 [48]. In the current study, treatment with old Δ130–136-PO CM significantly increased the expression of Col I and TGFβ1 in C2C12 myoblasts, and this increase was effectively inhibited after PGE2 supplementation in the aged Δ130–136-PO CM, which was further confirmed TGFβ/smad2/3 signaling was activated in vitro. Additionally, a recent work by Palla and colleagues [49] has revealed that inhibition of PGE2-degrading enzyme 15-PGDH augments aged muscle mass and strength, which provides additional support for the positive role of PGE2 signaling in maintaining muscle function. Collectively, these observations suggest impairment of osteocytic HCs inhibits PGE2 release and this reduction promotes collagen deposition in aging muscle via activation of TGFβ/smad2/3 signaling, leading to decreased muscle contractile force in old Δ130–136 mice.

In addition to fibrogenic phenotypes observed in Δ130–136 mice, mitochondrial homeostasis is another potential factor affecting muscle function in aged skeletal muscles [49]. Mitochondrial homeostasis in muscle tissues is maintained through biogenesis (Sirt-1) as well as the balance of fusion (OPA-1) and fission (Drp-1) activity [50]. In addition, ROS is required for normal mitochondria function [2], however, excessive ROS results in oxidative damage to mitochondrial DNA (mtDNA), lipids and proteins of skeletal muscle, which in turn leads to mitochondria breakdown, which adversely affects muscle function [51,52]. In contrast, antioxidant system comprised of CAT, SOD, and GSH can protect cells from damage caused by excessive ROS. In the present study, impairment of Cx43 HCs reduced the expression of OPA-1 and Sirt-1, while impairment of Cx43 GJs reduced Drp-1 expression in SL muscle of aged mice, indicating the impairment of Cx43 channels disrupts mitochondrial integrity. Additionally, we showed here that impairment of GJs in osteocytes increased ROS level accompanied by a decrease in activity of these endogenous antioxidants. Similarly, Le et al. [53] found that the absence of Cx43 GJs in astrocytes increased the ROS level induced by H_2_O_2_. However, a recent study by Jiang’s group [54] showed that Cx43 HCs prevent lens epithelial cells from oxidative stress through regulation of the ROS and GSH levels. These inconsistent results may reflect different roles of GJs and HCs in the regulation of ROS level, especially in different cell types. Nevertheless, the elevated ROS level in aging muscles provides another explanation for the decreased muscle contractile force in old Δ130–136 mice. However, the relationship between impaired Cx43 channels and increased ROS level in aged transgenic mice needs to be further investigated in our continuing studies.

Our transgenic models with overexpression of dominant-negative mutants offer a unique approach to dissect the specific involvement of these two channels formed by Cx43 [16]. In particular, the 10-kb Dmp1 promoter has been widely used to drive gene expression mainly in osteocytes [55,56]. In this study, the GFP-labeled mutant Cx43 protein was only expressed in osteocytes, not in other bone cells or adjacent muscle cells. However, other studies using the 10 kb-Dmp1 promoter with fluorescent probes target gene expression in osteocytes as well as skeletal muscle cells [57]. This discordance may be explained, at least in part, by the difference in sensitivity of the fluorescent probe used. Some of hypersensitive fluorescence signals may make it difficult to distinguish the difference in expression levels. Alternatively, Dmp1 promoter activity is not specific to a single cell type [25]. Nevertheless, our results revealed expression of the transgenes driven by Dmp1 promoter chiefly localized in osteocytes.

The present study has some limitations. First, PGE2 might be an essential mediator for communication between osteocytes and myoblasts, however, the role of PGE2 needs to be further illustrated in old mice in vivo. Second, only 10- and 18-month-old mice were used in this study. Likely older mice at age of 24 or 26 months would have been needed to further assess the effects of aging or sarcopenia in this model. Such expanded research will contribute to a comprehensive understanding of the involvement of Cx43 channels in bone-muscle crosstalk.

## 4. Materials and Methods

### 4.1. Transgenic Mice

Two transgenic mouse models overexpressing Cx43 dominant-negative mutants in osteocytes, R76W and Δ130–136, were generated as previously described [23]. Briefly, the DNA constructs used to generate these two Cx43 mutants harbor 10 kb Dmp1 promoter and exon1 (E1) of Dmp1, which are linked to Cx43 negative mutants and green fluorescent protein (GFP) with an intron. The specificity of the GFP-labeled Dmp-1 promoter in these two Cx43 mutants was evaluated by histological sections. As shown in Appendix A, the expression of GFP was predominantly detected in osteocytes but not surrounding muscle tissues. The 10- and 18-month-old transgenic male mice used in this study were housed in a temperature-controlled room with a light/dark cycle of 12 h at the Animal Research Lab of Northwestern Polytechnical University (NPU), under specific pathogen-free conditions. Food and water were freely available to the mice. Mice at 10 months of age correspond to humans at ~40 years of age and at 18 months of age correspond to ~60 years of age in humans [58]. Genotyping was performed by qPCR using genomic DNA extracted from mouse toes. All animal protocols were approved by the NPU Institutional Animal Care and Use Committee.

### 4.2. Micro-Computed Tomography (μCT) Analysis

Micro-CT scanning was performed as previously described [25]. Briefly, the left femur was isolated, cleaned of soft tissues, wrapped in saline-soaked gauze, and kept at −20 °C until experiments. The femur was scanned at 8μm resolution on a Micro-CT system (GE Healthcare, Madison, WI, United States) following standard parameters: 80 kV, 80 μA, rotation angle 360°, rotation angle increment 0.5°, exposure time 3000 s, and frame average 3. In the distal femoral metaphysis, the region of interest (ROI) of trabecular bone was positioned 0.5 mm proximal to the growth plate and extended proximally 1 mm. Trabecular structural parameters included: trabecular thickness (Tb.Th, mm), trabecular number (Tb.N, 1/mm), bone volume fracture (BV/TV, %), and trabecular separation (Tb.Sp, mm). Cortical bone structure was analyzed on a 0.5 mm-thick slice in the femoral midshaft (55% of length from the proximal to distal). Cortical parameters included: cortical area (Ct.Ar, mm^2^), cortical thickness (Ct.Th, mm), marrow area (Ma.Ar, mm^2^) and diaphyseal total area (T.Ar, mm^2^), bone area (B.Ar, mm^2^). All parameters were calculated according to published guidelines [59].

### 4.3. Histology and Immunohistochemistry

At sacrifice, the left tibia, isolated from transgenic mice and their WT littermates, was fixed in 4% (*w*/*v*) paraformaldehyde for 48 h prior to decalcification with 10% (*w*/*v*) ethylene diamine tetraacetic acid (EDTA) (pH = 7.2–7.4) for 4 weeks. The whole tibia bones then were embedded in paraffin and sagittal serial sections (5μm thickness) were prepared using a semi-automated rotary microtome (RM2235, Leica, Wetzlar, Germany). The tissue sections were either underwent hematoxylin and eosin (H&E) staining or were subjected to tartrate-resistant acid phosphatase (TRAP) staining according to our previously published protocols [25]. At least three bone sections per tibia were cut longitudinally as close to the center of bone as possible and photographed by the microscope (Nikon, 80i, Tokyo, Japan). The number of osteocytes (N.Ot), number of empty lacunae, number of endocortical osteoblasts (N.Ob), number of endocortical osteoclasts (N.Oc), bone area (B.Ar), and bone surface area (BS) were quantified by Image J software (National Institutes of Health, Bethesda, MD, USA). N.Ot/B.Ar, N.Ob/BS, N.Oc/BS and Oc.S/BS ratio also were calculated, respectively.

Tibia paraffin sections were also used for immunohistochemistry as previously reported [25]. Briefly, sections were deparaffinized, treated by citric acid antigen retrieval buffer (Servicebio, Wuhan, China) for 8 min in a microwave for antigen retrieval. After washing, the sections were treated with 3% (*v*/*v*) hydrogen peroxide solution for 25 min at room temperature to block intrinsic peroxidase activity. Sections then were blocked with 3% (*v*/*v*) Bovine Serum Albumin (BSA) (Salarbio, Beijing, China; A8020) for 30 min at room temperature. Sections were incubated overnight at 4 °C with primary antibodies against active caspase-3 (Servicebio; GB11572, 1:500). Following three washes in PBS, the sections were incubated with horseradish peroxidase (HRP)-labeled goat anti-rabbit secondary antibody (Servicebio, Wuhan, China) for 50 min at temperature and followed by color reaction using diaminobenzidine (DAB, G1211, Servicebio) and hematoxylin counterstain. The bone slides were photographed with the microscope (Nikon, 80i, Tokyo, Japan) and images were analyzed by Image J software (NIH, Bethesda, MD, USA).

Soleus (SL) isolated from mice were fixed in 4% (*w*/*v*) paraformaldehyde for 48 h, subsequently embedded in paraffin and sliced for serial 5μm-thick cross-sections from the midbelly of SL muscles using a semi-automated rotary microtome (RM2235, Leica, Wetzlar, Germany). The sections were then stained with hematoxylin eosin (H&E) and Sirius red. Myofiber cross-sectional area (CSA) and myofiber number by H&E staining and collagen area (%) by Sirius red staining were then quantified, respectively. To detect the fiber type composition in SL muscles, immunohistochemistry was performed in paraffin sections using anti-skeletal slow myosin antibody (Mouse mAb, 1:2000; Sigma–Aldrich, St. Louis, MO, USA) with HRP-labelled goat anti-rabbit secondary antibody (Servicebio, Wuhan, China) and followed by color reaction using diaminobenzidine (DAB, G1211, Servicebio, Wuhan, China) and hematoxylin counterstain. Muscle slides were photographed by the microscope (Nikon, 80i, Tokyo, Japan) and percentage of type I myofibers in each muscle were calculated by Image J software (NIH, Bethesda, MD, USA).

### 4.4. Ex Vivo Muscle Function Testing

Ex vivo muscle contractile force was measured as previously described [60]. Briefly, mice were anesthetized by with 5% (*ν*/*ν*) vaporized Isoflurane (RWD, R510-22, Shenzhen, China) mixed with O_2_. The SL muscle was isolated from the mice while maintaining its innervation and normal blood supply. The distal tendon of the SL muscle was tied to a MLT0420 force transducer (Panlab, Sydney, Australia) with surgical suture, and the proximal tendon of the muscle was secured to a fixed steel post. Electrical stimulation output from the stimulator (SEN-3301, Tokyo, Japan) was applied to the muscle via the two electrodes placed on the muscle. The maximal absolute twitch tension was measured by the single square-wave pulses with maximal stimulation voltage 5 V and 25 ms in duration at intervals of 2 s. The maximal absolute tetanic tension was measured by the repeated square-wave multi-pulse with maximal stimulation voltage 5 V and 25 ms in duration at intervals of 40 ms. After force measurement, we calculated the specific force by normalizing the absolute force measurements by the muscle CSA, which was measured according to the published protocols [61].

### 4.5. Western Blotting Analysis

Cell samples were lysed on ice in RIPA Lysis Buffer supplemented with a Protease and Phosphatase Inhibitor Cocktail (CWBIO, Beijing, China). The SL muscle samples in ice-cold RIPA Lysis buffer (Beyotime, P0013B, Haimen, China) containing a Protease and Phosphatase Inhibitor Cocktail (CWBIO, Beijing, China) were homogenized using a tissue homogenizer (Servicebio, KZ-II, Wuhan, China). All lysates were then centrifuged at 12,000× *g* at 4 °C for 15 min. The supernatants were collected, and protein concentration was quantified using a BCA Protein Assay kit (Beyotime, P0010, Shanghai, China). Protein samples were then mixed with 5× loading buffer (Fudebio-tech, FD002, Hangzhou, China), and then denatured at 100 °C for 5 min. An equal amount of protein (30 μg) was separated on 8% or 10% SDS-PAGE gels (100 V, 120 min) and then electrophoretically transferred onto PVDF membrane (IPVH00010, Merck Millipore, Burlington, MA, USA). After transfer, the membranes were blocked in the 5% (*w*/*ν*) non-fat milk powder (BD Biosciences, 232100, Franklin Lakes, NJ, USA) for 2 h at room temperature. The membranes were probed with specific primary antibodies against collagenI (rabbit mAb, 1:1000; Abclone, A5786, Wuhan, China), TGF-β1 (rabbit mAb, 1:1000; Abclone, A2124, Wuhan, China), Smad2/3 (rabbit mAb, 1:1000; Beyotime, AF8001, Shanghai, China), p-Smad2/3 (rabbit mAb, 1:1000; Abclone, AP0548, Wuhan, China), OPA1 (rabbit mAb, 1:1000; Beyotime, AF7653, Shanghai, China), Drp1 (rabbit mAb, 1:1000; Beyotime, AF6735, Shanghai, China), Sirt-1 (rabbit mAb, 1:1000; Beyotime, AF0282, Shanghai, China), and GAPDH (rabbit mAb, 1:1000; Beyotime, AF1186, Shanghai, China) at 4 °C overnight. Subsequently, membranes were incubated with horseradish peroxidase (HRP)-linked secondary antibody (1:2000; Beyotime, Shanghai, China) for 2 h at room temperature. After washing three times, the membranes overlaid with chemiluminescence (ECL) regents were visualized using a chemiluminescence detection system (T5200Multi; Shanghai, China). The gray-scale densities of the protein bands were analyzed using ImageJ (NIH) software.

### 4.6. Primary Osteocytes Isolation, Conditioned Media Preparation, and Cell Cultures

Primary osteocytes were isolated from marrow-flashed long bones of 18-month-old trans genic mice and their WT littermates, as previously described [23]. Briefly, long bones were dissected and sequentially digested with digestion solution containing 0.15% (*v*/*v*) collagenase type I (BD Biosciences, Concord, MA, USA) and EDTA (5 mM) in an incubator under 37 °C and 5% CO_2_. The cells isolated from the first three digestions were removed and the final digests enriched osteocytes were cultured in alpha Modified Eagle’s Medium (α-MEM, Gibco, Carlsbad, CA, USA) supplemented with 5% fetal bovine serum (FBS, Hyclone, Logan, UT, USA), 100 U/mL penicillin and 100 μg/mL streptomycin (Beyotime, Shanghai, China). After 48 h, the CM was collected and stored at −80 °C until use. The C2C12 myoblasts were cultured in high-glucose Dulbecco’s modified Eagle’s medium (DMEM, GIBCO, Carlsbad, CA, USA) containing 10% (*ν*/*ν*) fetal bovine serum (FBS, Hyclone, Logan, UT, USA), 100 U/mL penicillin and 100 μg/mL streptomycin (Beyotime, Haimen, China) for 48 h. All cell lines were cultured under 5% CO_2_ and 37 °C in a controlled humidified incubator.

### 4.7. RNA Extraction and Real-Time PCR (qPCR)

The muscle and marrow-flushed tibia of mice were harvested, frozen in liquid nitrogen and stored at −80 °C. The total RNA was extracted from tissues or cultural cells using TRIzol Regent (Invitrogen, Carlsbad, CA, USA). One microgram of total RNA was reversely transcribed using HiScript II Q RT supermix for qPCR (Vazyme, Nanjing, JiangSu, China; R223-01). Real-time PCR analysis was performed with cham Q SYBR qPCR Master Mix according to the manufacturer’s instructions (Vazyme; Q311-01). GAPDH was used as a housekeeping gene control. The expression of target genes was calculated by the 2^−ΔΔCt^ method using the WT or WT-PO CM control group as the second control. All primers in this study were synthesized by Sangon Biotech (Shanghai, China) and the sequences were shown in Appendix A.

### 4.8. Co-Treatment of C2C12 Cells with Δ130–136 PO CM and PGE_2_

The C2C12 cells were seeded on 24-well plates and cultured for 24 h, after which media was replaced with differentiation media (DM) containing 2.5% (*ν*/*ν*) horse serum (GIBCO, Carlsbad, CA, USA),100 U/mL penicillin and 100 μg/mL streptomycin in DMEM with 25% Δ130–136 PO CM. The C2C12 cells were incubated in the absence or presence of 1 μM PGE2 (Aladdin, D133402, Shanghai, China) for 72 h, and the protein samples were collected. The dose of exogenous PGE2 supplementation was based on previous studies [62,63], with minor modifications.

### 4.9. ROS Level and Antioxidant Enzyme Activity

The SL muscles of mice were harvested, weighed, and homogenized using a tissue homogenizer (Servicebio, KZ-II, Wuhan, China). Reactive oxygen species (ROS) level and activities of antioxidant enzymes including catalase (CAT), superoxide dismutase (SOD) and glutathione (GSH) in SL muscle samples were then determined according to manufacturer’s instructions (Jianglai Biotech; Shanghai, China). The ROS level and antioxidant enzyme activity were normalized to their respective tissue weights.

### 4.10. Biochemical Assay

The biochemical assays were performed on both blood samples and primary osteocyte conditioned media (PO CM). Blood was collected from the heart of anesthetized mice after fasting overnight. The serum was fractionated after the blood samples were centrifuged as previously described [64]. Serum adenosine triphosphate (ATP) and prostaglandin E2 (PGE2) in PO CM were determined using the mouse enzyme linked immunosorbent assay (ELISA) kit (Jianglai Biotech; Shanghai, China; Abcam, Cambridge, UK) according to the manufacturer’s instructions, respectively.

### 4.11. Statistical Analysis

Data are presented as mean ± SD. The statistical analysis was performed by the Prism 6.0 software (GraphPad, La Jolla, CA, USA). Data distributions were compatible with ANOVA analysis. One-way ANOVA with Tukey’s multiple comparisons test was performed to assess difference between WT and age-mated transgenic mice (* *p* < 0.05; ** *p* < 0.01).

## 5. Conclusions

Our findings indicate that the enhancement of Cx43 HCs while GJs are inhibited in osteocytes decreases bone mass in aged male mice. Moreover, impairment of Cx43 HCs inhibits PGE2 level in osteocytes and this reduction promotes collagen deposition in aging muscle through activation of TGFβ/smad2/3 signaling, which together with increased ROS level contributes to decreased muscle contractile force in aged male mice. Understanding the role of osteocytic Cx43 channels in bone and muscle with aging is beneficial to provide potential novel therapies for the twin aging-diseases, osteoporosis, and sarcopenia.

## Figures and Tables

**Figure 1 ijms-23-13506-f001:**
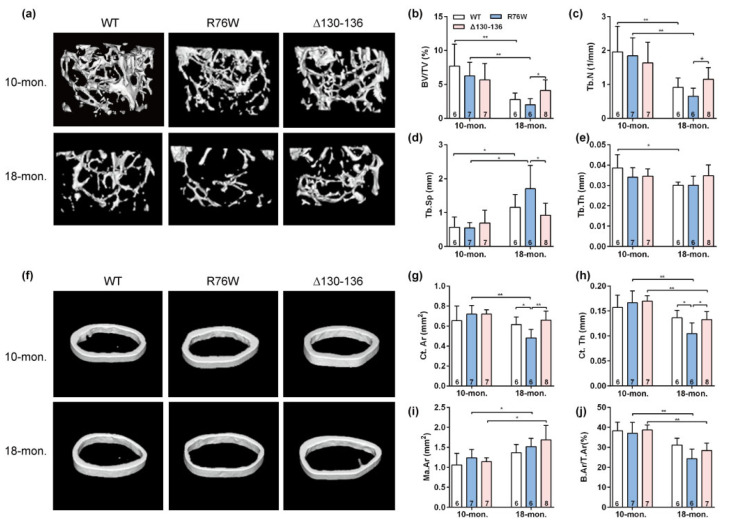
Enhancement of Cx43 HCs while GJs are inhibited decreases bone mass in 18-month-old mice. (**a**) Representative μCT images of femoral trabecular microarchitecture. (**b**–**e**) Trabecular microstructural parameters, including BV/TV (**b**), Tb.N (**c**), Tb.Sp (**d**) and Tb.Th (**e**), were shown. (**f**) Representative μCT images of femoral cortical microarchitecture. (Jung-Jun, Berggren, Hulver, Houmard, and Hoffman) Cortical microstructural parameters at the midshaft femur, including Ct.Ar (**g**), Ct.Th (**h**) Ma.Ar (**i**) and B.Ar/T.Ar (**j**) were shown. Sample sizes are indicated at the bottom of each column. BV/TV = bone volume fraction; TB.N = trabecular thickness; Tb.Sp = trabecular separation; Tb.Th = trabecular thickness; Ct.Ar = cortical area; Ct.Th = cortical thickness; Ma.Ar = marrow area; B.Ar = bone area; T.Ar = total area. * *p* < 0.05; ** *p* < 0.01.

**Figure 2 ijms-23-13506-f002:**
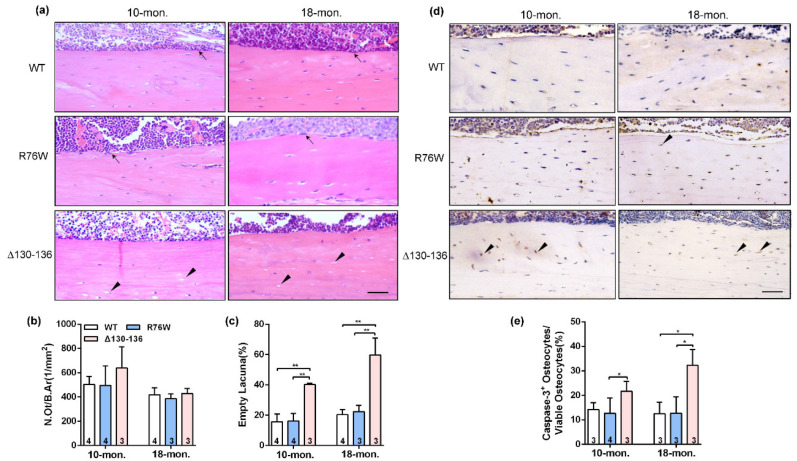
Significant increase of apoptotic osteocytes and empty lacunae in Δ130–136 mice. (**a**) Representative H&E staining images of tibial cortical bone in 10-and 18-month-old transgenic mice and their littermates. The black solid arrowheads indicate empty lacunae and arrows represent osteoblasts on the bone surface. Scale bar = 50 μm. (**b**) The number of osteocytes was assessed by N.Ot/B.Ar. (**c**) The percentage of apoptotic osteocytes with total osteocytes. (**d**) Representative immunohistochemical images of active Caspase-3 in mid-diaphyseal cortical bone in 10- and 18-month-old transgenic mice and their littermates. The black solid arrowheads indicate empty lacunae active Caspase-3-positive stained osteocytes. Scale bar = 50 μm. (**e**) The percentage of caspase-3-positive osteocytes with total osteocytes. Sample sizes are indicated at the bottom of each column. * *p* < 0.05; ** *p* < 0.01.

**Figure 3 ijms-23-13506-f003:**
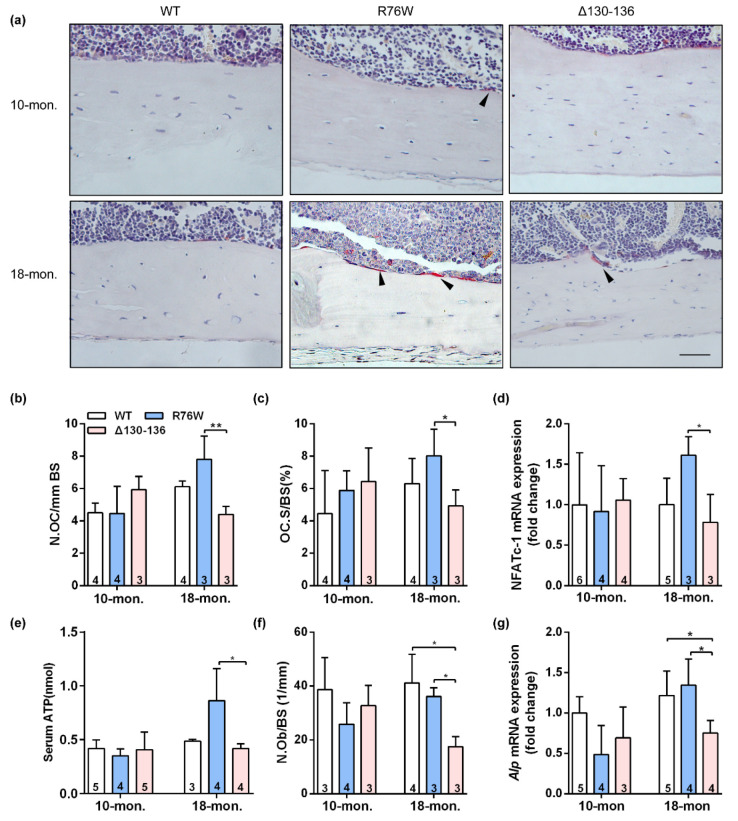
Effects of Cx43 channels on endocortical osteoclasts number and surfaces in transgenic mice. (**a**) Representative TRAP staining images of tibial cortical bone 10- and 18-month-old transgenic mice and their littermates. The black arrows indicate osteoclasts on bone surface. Scale bar = 50 μm. (**b**,**c**) The number and surface of osteoclasts were estimated by N.Oc/BS (**b**) and Oc.S/BS (**c**), respectively. (**d**) Real-time PCR analysis of mRNA expression of NFATc-1 in tibia. (**e**) Serum concentration of ATP by ELISA assay in mice. (**f**) The number of osteoblasts were evaluated by N.Ob/BS. (**g**) Real-time PCR analysis of Alp in tibia. Sample sizes are indicated at the bottom of each column. * *p* < 0.05; ** *p* < 0.01.

**Figure 4 ijms-23-13506-f004:**
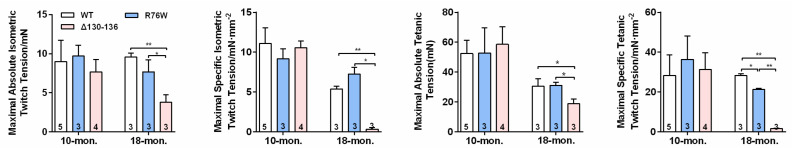
Impairment of Cx43 HCs decreases SL muscle contractile force in 18-month-old mice. (**a**–**d**) Maximal absolute isometric twitch tension (**a**), maximal specific isometric twitch tension (**b**), maximal absolute tetanic tension (**c**) and maximal specific tetanic tension (**d**) in SL muscles in 10-and 18-month-old transgenic mice and their WT littermates. Sample sizes are indicated at the bottom of each column. * *p* < 0.05; ** *p* < 0.01.

**Figure 5 ijms-23-13506-f005:**
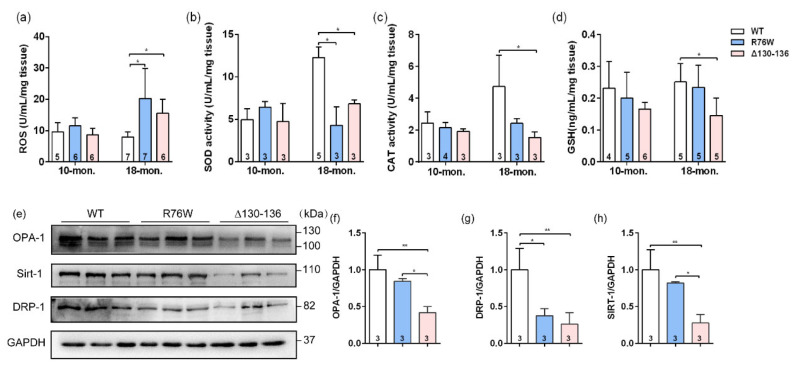
Effects of Cx43 channels on oxidative stress and mitochondrial homoeostasis in muscles of 18-month-old transgenic mice. (**a**–**d**) Levels of ROS (**a**), SOD activity (**b**), CAT activity (**c**) and GSH (**d**) in muscle of aged transgenic mice and their WT littermates. (**e**) Western blotting images of marker proteins related to muscle mitochondrial homoeostasis; (**f**–**g**) Quantifications of protein expression for OPA-1 (**f**), Drp-1 (**g**) and Sirt-1 (**h**) in (**e**). Sample sizes are indicated at the bottom of each column. * *p* < 0.05; ** *p* < 0.01.

**Figure 6 ijms-23-13506-f006:**
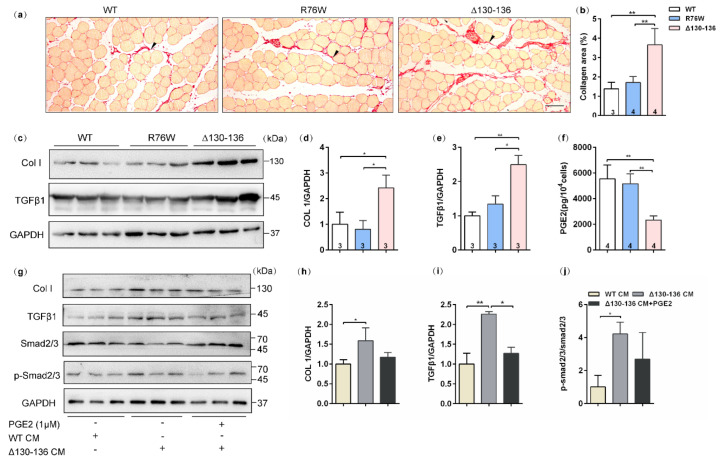
Increased muscle fibrotic phenotype in 18-month-old Δ130–136 mice and exogenous PGE_2_ partially prevented collagen deposition. (**a**) Representative Sirius staining images of SL muscles in aged transgenic mice and their WT littermates. The solid arrowheads indicate deposited collagen. Scale bar = 50 μm. (**b**) Quantification of collagen content in SL muscles of aged mice. (**c**) Western blotting images of Collagen I (Col I) and TGFβ1 in SL muscles. (**d**,**e**) Quantifications of protein expression for Col I (**d**) and TGFβ1 (**e**) in (**c**). (**f**) The concentration of PGE_2_ in primary osteocyte conditioned media (PO CM) from 18-month-old transgenic mice and their WT littermates. (**g**) Western blotting images of expression of proteins related to collagen deposition in the absence or presence exogenous PGE_2_. (**h**–**j**) Quantifications of protein expression in (**g**). Sample sizes are indicated at the bottom of each column. * *p* < 0.05; ** *p* < 0.01.

## Data Availability

Data are contained within the article.

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
