# Peer review of "Connexin 43 Channels in Osteocytes Are Necessary for Bone Mass and Skeletal Muscle Function in Aged Male Mice"

_ijms, 2022, doi:10.3390/ijms232113506_

Round 1

Reviewer 1 Report

Overall comments: The comprehensive study of Li et al. describes the consequences of overexpressing Cx43 mutants on bone and muscle characteristics in 10 and 18-month-old mice to dissect out the different roles of Cx43-formed gap junction channel (GJs) and hemichannels (HCs) in bone-muscle crosstalk during aging. The large array of in vivo and in vitro results provide new experimental data, in addition to the data from their group’s previous observations on younger animals. The authors conclude that enhancement of Cx43 HCs in the absence of gap junction activity reduces bone mass in aged mice, whereas inhibition of the hemichannels leads to a reduction in PGE2 release in osteocytes, increase muscle collagen synthesis through activation of TGFβ/smad2/3 signaling, which together with high ROS level leads to lower muscle force in aged mice. The study is of interest and the data enhance our understanding of bone-to-muscle crosstalk. In addition, the experiments were well designed and data analysis and conclusion were sound. The manuscript could be further improved by addressing some minor issues as outlined below.

Minor/Stylistic Comments:

1pg1. line 33: The current statement is not clear. Please rephrase the entire sentence.

2pg1. Line29-39: The results and conclusions need to be condensed further in the abstract section.

32.1.: The title “Enhancement of HCs while GJs Is Inhibited Reduced Bone Mass in Older Mice” seems to have grammatical error. Please modify the statement in an appropriate fashion.

4line 380: The two transgenic mice models used in the study harbor 10kb Dmp1 promoter and the authors also presented supplemental figure 1 to confirm the specificity of the DMP1 in osteocytes. Are there still other alternative methods to examine its specificity?

5Line 490: Was the bone marrow included in the preparations used to isolate osteocytes?

6Line 492:What the percentage of isolated primary osteocyte in this experiments?

7It is important to justify the gender (male mice) and the age (10 and 18 months) for the study. Based on previous studies in aging field, the authors choose 10- and 18-month-old mice corresponding to adult and old people. How does this relate to physiologic bone loss in humans?

8The sex of the mice should be indicated in the conclusion section.

9The full names of the abbreviations in the manuscript need to be provided to better understand what it means.

10The surfaces in which cells were counted should be indicated in the methods and results section (cancellous, endocortical, periosteal?).

11There are some typos and grammatical mistakes throughout the manuscript. Attention should be made to the presentation of the manuscript.

12Do the authors mean the differences in Myh7 mRNA levels in supplementary material were not significant? What do Myh7 mRNA mean in the manuscript?

13Why did the author choose to concentrate on SL muscle in their study? Does this muscle contain only slow-twiching muscle fibres?

14Please rephrase the statement in Line 150.

15line390: Estrogens and androgens both have profound effects in bone and skeletal muscle. Therefore, the justification for only using male mice is not appropriate and should be removed

Author Response

Dear editor  and reviewer

We very much appreciate the positive review of our paper, and constructive suggestions and comments by the reviewers. We have marked the changes in red in the revised manuscript.

Reviewer’s comments: The comprehensive study of Li et al. describes the consequences of overexpressing Cx43 mutants on bone and muscle characteristics in 10 and 18-month-old mice to dissect out the different roles of Cx43-formed gap junction channel (GJs) and hemichannels (HCs) in bone-muscle crosstalk during aging. The large array of in vivo and in vitro results provide new experimental data, in addition to the data from their group’s previous observations on younger animals. The authors conclude that enhancement of Cx43 HCs in the absence of gap junction activity reduces bone mass in aged mice, whereas inhibition of the hemichannels leads to a reduction in PGE2 release in osteocytes, increase muscle collagen synthesis through activation of TGFβ/smad2/3 signaling, which together with high ROS level leads to lower muscle force in aged mice. The study is of interest and the data enhance our understanding of bone-to-muscle crosstalk. In addition, the experiments were well designed and data analysis and conclusion were sound. The manuscript could be further improved by addressing some minor issues as outlined below.

Response: Thanks for your positive comments on our paper.

Minor/Stylistic Comments:

1pg1. line 33: The current statement is not clear. Please rephrase the entire sentence.

Response: Thanks for your suggestion. We have rephrased the sentence in line 33.

2pg1. Line29-39: The results and conclusions need to be condensed further in the abstract section.

Response: Thanks for your suggestion. We have condensed the abstract in the text.

32.1.: The title “Enhancement of HCs while GJs Is Inhibited Reduced Bone Mass in Older Mice” seems to have grammatical error. Please modify the statement in an appropriate fashion.

Response: We are sorry to the mistake. The grammatical error in the sentence has been revised.

4line 380: The two transgenic mice models used in the study harbor 10kb Dmp1 promoter and the authors also presented supplemental figure 1 to confirm the specificity of the DMP1 in osteocytes. Are there still other alternative methods to examine its specificity?

Response: In previous studies, multiple promoter-Cre transgenic mice have been generated to examine the specificity of Dmp1 promoter in osteocyte. Lu et al. utilized 10-kb Dmp1 promoter to direct expression of Cre recombinase. A different reporter model in which a recombination event causes activation of tdTomato fluorescent protein was used to evaluate the Cre activity. Another approach to detect the specificity of Dmp1 is to use a tamoxifen-inducible Cre system (CreERT2) in which a mutated estrogen ligand binding domain fusing to the Cre recombinase.

Lu Y, Xie Y, Zhang S, Dusevich V, Bonewald LF, Feng JQ. DMP1-targeted Cre expression in odontoblasts and osteocytes. J Dent Res 2007;86:320-3255.

Powell Jr WF, Barry KJ, Tulum I, Kobayashi T, Harris SE, Bringhurst FR, et al. Targeted ablation of the PTH/PTHrP receptor in osteocytes impairs bone structure and homeostatic calcemic responses. J Endocrinol 2011;209:21-32

5Line 490: Was the bone marrow included in the preparations used to isolate osteocytes?

Response: Bone marrow was not included in the preparations used to isolate osteocytes. We have rephrased the statement in the text.

6Line 492:What the percentage of isolated primary osteocyte in this experiments?

Response: In vitro, the percentage of isolated primary osteocyte from 10-and 18-month-old transgenic mice is 75-80%.

7It is important to justify the gender (male mice) and the age (10 and 18 months) for the study. Based on previous studies in aging field, the authors choose 10- and 18-month-old mice corresponding to adult and old people. How does this relate to physiologic bone loss in humans?

Response: Thanks for your comment. According to Jackson Lab’s criteria for the age of mice, we used 10 and 18-month-old transgenic mice in the study. Adult mice should be at least 10 months old, an age equivalent to middle-aged humans. For the old-aged group, mice should be at least 18 months old, an age equivalent to old age in humans (Flurkey et al. 2007). In general, bone loss starts soon after age 30, but become a rapid, progressive, debilitating condition after age 60 (Bonewald 2018). Therefore, mice at age of 10 months and 18 months are useful animal models to study underlying mechanisms related to physiologic bone loss in humans.  

Flurkey, K., Currer, J. M., & Harrison, D. E. (2007). Mouse models in aging research. In The mouse in biomedical research (pp. 637-672). Academic Press.

Bonewald, L. (2019). Use it or lose it to age: A review of bone and muscle communication. Bone, 120, 212-218.

8The sex of the mice should be indicated in the conclusion section.

Response: Thanks for your suggestion. We have added the sex of the mice in the conclusion section.

9The full names of the abbreviations in the manuscript need to be provided to better understand what it means.

Response: Thanks for your suggestion. We have added the abbreviations in the text.

10The surfaces in which cells were counted should be indicated in the methods and results section (cancellous, endocortical, periosteal?).

Response: Thanks for your suggestion. We have revised the statement in the text.

11There are some typos and grammatical mistakes throughout the manuscript. Attention should be made to the presentation of the manuscript.

Response: Thanks for your comment. We have revised some typos and grammatical errors throughout the text.

12Do the authors mean the differences in Myh7 mRNA levels in supplementary material were not significant? What do Myh7 mRNA mean in the manuscript?

Response: Thanks for your comment. The mRNA levels were not significant across groups in mice at age of 10 or 18 months. The gene, Myh7, encodes MyHC I protein that compose slow-twitching muscle fibers.

13Why did the author choose to concentrate on SL muscle in their study? Does this muscle contain only slow-twiching muscle fibres?

Response: In our study, we also examined the phenotypes of gastrocnemius, a fast-twitch muscle (data not shown). However, we did not observe obvious changes in muscle mass or contractile force in transgenic mice at either 10 or 18 months of age. Therefore, we focused on the soleus muscle, which mainly consists of slow-twitch fibers.

14Please rephrase the statement in Line 150.

Response: Thanks for your comment. We have revised the statement in Line 150.

15line390: Estrogens and androgens both have profound effects in bone and skeletal muscle. Therefore, the justification for only using male mice is not appropriate and should be removed

Response: Thanks for your comment. We have removed the justification for only using male mice in the material and methods. 

Reviewer 2 Report

The authors want to dissect the role of connexin 43-formed gap junctions and hemichannels in osteocytes during aging. For this purpose, they used two different mouse models that have already been published, in which connexin 43 is mutated only in gap junctions (R76W) leading to an inhibition of these channels or in both gap junctions and hemichannels (D130-136). From their experiments, they said that Cx43 hemichannels inhibit PGE2 in osteocytes, leading to increased collagen deposition in skeletal muscle, and thus to muscle weakness. They speculated about the function of Cx43 hemichannels comparing the D130-136 mutant, where both channels are blocked, with the R76W mutant, where only gap junctions are inhibited.

However, in the latter model, it has been published that gap junction inhibition occurs together with HCs enhancement. For this reason, it is difficult to understand if the effects that occurred in D130-136 mutant are only related to HCs block or to a synergistic effect of the double inhibition. A model in which gap junctions are blocked with no alterations in HCs could potentially address this point.

My main concern is about the aging model. The authors highlight the importance of studying both osteoporosis and sarcopenia as twin-aging diseases. However, no differences have been observed between 10-month and 18-month-old mice regarding both bone and muscle parameters. I would maybe suggest using as control an adult mouse, thus younger compared to 10-month-old group, in order to better dissect the role of Cx43 channels during the aging process. In addition, the authors concluded that GJs inhibition and HCs enhancement lead to decreased bone mass and increased osteoclast number even if there are no significant differences with wildtype animals.

- Figure 1: according to the previous concern, the image in Fig.1a shows major differences in trabecular architecture between 10-month and 18-month and in aged group between wt and R76W model. However, this is not reflected by the histogram in Fig.1b-e, where no changes are observed between the two timepoints or when compared to wt. The authors should increase the number of mice or change the images according to the histogram results.

- Figure 4 and relative materials and methods section: The way in which the authors performed ex vivo muscle force is not clear. They put a reference in the materials and methods section (Blau et al., PNAS 2017) where another protocol is used. The authors said that they maintain muscle innervation (line 453) and, also that they placed two electrodes on the muscle (line 457). To me, it is not clear if they are directly stimulating the muscle or their nerve. In addition, usually for ex vivo muscle measurements, the duration of the pulses varies from 500us to 1ms. The authors are using 25ms, which is a very long duration time for a stimulus. Moreover, for the tetanic stimulation, they are using 40ms of intervals, which should correspond to 25Hz, which is not the correct frequency to induce tetanus. Did the authors place the soleus muscle in a bath with a controlled temperature or not? This is because also the temperature is affecting picks fusion during tetanus. Considering this, the authors should perform again the experiment with a correct protocol (Delavar H, Nogueira L, Wagner PD, Hogan MC, Metzger D, Breen EC. Skeletal myofiber VEGF is essential for the exercise training response in adult mice. Am J Physiol Regul Integr Comp Physiol. 2014 Apr 15;306(8):R586-95. doi: 10.1152/ajpregu.00522.2013. Epub 2014 Feb 12. PMID: 24523345; PMCID: PMC4043130).

Regarding the figure, maximal values are lower compared to published literature (this can be due to the protocol used). In addition, why does the sample size between the absolute force and the normalized one change? Are these analyses been performed on different samples? They should be the same. Also, the authors did not see differences in muscle weight/body weight and fiber CSA. However, the differences between wt, R76W, and D130-136 mutant are exacerbated when absolute force is normalized on fiber area. Why this huge difference?

- Paragraph 2.6: The authors said that Cx43 channels impairment alters mitochondrial function. However, this is a hypothesis based on ROS content and expression of proteins involved in mitochondrial dynamics. They do not perform any assay for the analysis of mitochondrial functionality. The authors should measure membrane potential through TMRM and mitochondrial respiration. And they should include also adult mice as control.

In addition, the authors conclude that mitochondrial impairment and ROS content can be responsible for muscle weakness. But, ROS levels are higher also in R76W mutant where muscle force is not affected. Do the authors have an explanation for this? Moreover, would an anti-oxidant treatment be able to restore muscle force to normal levels in D130-136 mutant?

- It is not clear to me why the authors decided to put the section on in vitro C2C12 differentiation with osteocyte CM, also because the effect that they saw on myotube diameter is not in line with CSA measurements. Considering that the 2 models and analysis cannot be compared and that this experiment does not add anything to the story, in my opinion, I would suggest removing it. In case, they decided to maintain it, I suggest putting more evident images of myotubes IF. 

 - In the abstract, the authors said that osteocyte apoptosis in aged mice is due to reduced PGE2 levels. This is speculation because they did not check if PGE2 supplementation leads to decreased osteocyte apoptosis since they performed the experiment only in C2C12 myotubes. Also, the authors concluded that the decreased PGE2 levels due to Cx43 impairment are responsible for muscle fibrosis in aged mice. But, did the authors check the levels of PGE2 in the circulation of the 3 mouse models in young and aged mice? Is it possible that PGE2 secreted from osteocytes is affecting other tissues, like muscle, in this case, considering that it is also produced by other organs? And if the authors find it to be decreased in the circulation, could a restoration of PGE2 to normal levels revert the fibrotic phenotype in aged muscles? And what about muscle weakness?

- PGE2 is also controlling muscle regeneration and MuSC function. Did the authors check for these parameters in their mice?

Minor points:

- a list of abbreviations should be added at the beginning of the manuscript or at least, the first time you write an abbreviation the authors should describe it. 

- in line with this, in Paragraph 2.3, the authors should explain what Alp is and why they decided to show NFAT-c, ATP and Alp in this context. 

- line 458: pulses not pluses

- line 150: maybe it is osteoclast number rather than osteoblast n. that is changing when GJs are inhibited

- Trap staining for osteoclast should be more evident. Images are not clear

- Graphs color should be changed: in the blue bars the sample size is difficult to read

- Figure S3 should be ameliorated, maybe leaving all the muscle attached to the bone and cutting them together 

Author Response

Dear editor and reviewer

We thank to the anonymous reviewer for your constructive comments and suggestions to improve the quality of the paper. We have carefully addressed the reviewers’ comments point by point carefully and marked the changes in red in the revised manuscript.  

  1. The authors want to dissect the role of connexin 43-formed gap junctions and hemichannels in osteocytes during aging. For this purpose, they used two different mouse models that have already been published, in which connexin 43 is mutated only in gap junctions (R76W) leading to an inhibition of these channels or in both gap junctions and hemichannels (D130-136). From their experiments, they said that Cx43 hemichannels inhibit PGE2 in osteocytes, leading to increased collagen deposition in skeletal muscle, and thus to muscle weakness. They speculated about the function of Cx43 hemichannels comparing the D130-136 mutant, where both channels are blocked, with the R76W mutant, where only gap junctions are inhibited.

However, in the latter model, it has been published that gap junction inhibition occurs together with HCs enhancement. For this reason, it is difficult to understand if the effects that occurred in D130-136 mutant are only related to HCs block or to a synergistic effect of the double inhibition. A model in which gap junctions are blocked with no alterations in HCs could potentially address this point.

Response 1: Thanks for your suggestion to improve the quality of our paper. According to our previously published paper, the channel function of gap junction and hemichannels in osteocytes isolated from transgenic mice was determined using dye coupling and dye uptake, respectively. EtBr uptake was increased in osteocytes from R76W and wild-type (WT) mice, not in Δ130‐136, in the presence of flow shear stress (FSS) (Xu et al. 2015). Thus, osteocyte expressing Δ130‐136 mutant has dominant effect both on gap junctions and hemichannels while R76W mutant only has dominant effect on gap junctions. Thus, our transgenic mouse models offer a unique approach to dissect the different roles of these two channels in bone and muscle phenotypes. 

[1] H. Xu, S. Gu, M.A. Riquelme, S. Burra, D. Callaway, H. Cheng, et al., Connexin 43 channels are essential for normal bone structure and osteocyte viability, J. Bone Miner. Res. 30 (2015): 436-448.

  1. My main concern is about the aging model. The authors highlight the importance of studying both osteoporosis and sarcopenia as twin-aging diseases. However, no differences have been observed between 10-month and 18-month-old mice regarding both bone and muscle parameters. I would maybe suggest using as control an adult mouse, thus younger compared to 10-month-old group, in order to better dissect the role of Cx43 channels during the aging process. In addition, the authors concluded that GJs inhibition and HCs enhancement lead to decreased bone mass and increased osteoclast number even if there are no significant differences with wildtype animals.

Response 2: Thanks for your suggestion and comment. It is indeed a good idea that a control younger mouse should be used to better dissect the role of Cx43 channels during the aging process. In our previous studies, using 4- and 3-month-old transgenic mice (young group), we have investigated the different roles of gap junctions and hemichannels in bone (Xu et al. 2015) and muscle phenotypes (Li et al. 2021), respectively. Therefore, transgenic mice at 10 and 18 months were selected to examine the functional contribution to aging bone and muscle in the present study. In addition, lower bone mass were observed in R76W mice compared to Δ130‐136 but R76W showed a non-significant decreasing trend (P>0.05) compared with WT groups, which may be attributed to small sample size of animals. Given that trabecular bone mass in aged R76W and Δ130-136 mice had no significant difference compared to WT littermates, this may suggest that combined effects of GJs inhibition and HCs enhancement, not GJs, were responsible for the low bone mass.

[1] H. Xu, S. Gu, M.A. Riquelme, S. Burra, D. Callaway, H. Cheng, et al., Connexin 43 channels are essential for normal bone structure and osteocyte viability, J. Bone Miner. Res. 30 (2015): 436-448.

[2] Li, G., Zhang, L., Ning, K., Yang, B., Acosta, F. M., Shang, P, et al., Osteocytic Connexin43 Channels Regulate Bone–Muscle Crosstalk. Cells. 2021.10(2), 237.

  1. Figure 1: according to the previous concern, the image in Fig.1a shows major differences in trabecular architecture between 10-month and 18-month and in aged group between wt and R76W model. However, this is not reflected by the histogram in Fig.1b-e, where no changes are observed between the two timepoints or when compared to wt. The authors should increase the number of mice or change the images according to the histogram results.

Response 3: Thanks for your suggestion. Since we mainly focused the differences in all three genotypes in 10 months or 18 months rather than the differences between the two time-points. We have changed the representative images according to the histogram results in Figure 1. Also, we also added the comparisons between two time-points in bone parameters and partial statements have been revised accordingly.

  1. Figure 4 and relative materials and methods section: The way in which the authors performed ex vivomuscle force is not clear. They put a reference in the materials and methods section (Blau et al., PNAS 2017) where another protocol is used. The authors said that they maintain muscle innervation (line 453) and, also that they placed two electrodes on the muscle (line 457). To me, it is not clear if they are directly stimulating the muscle or their nerve. In addition, usually for ex vivo muscle measurements, the duration of the pulses varies from 500us to 1ms. The authors are using 25ms, which is a very long duration time for a stimulus. Moreover, for the tetanic stimulation, they are using 40ms of intervals, which should correspond to 25Hz, which is not the correct frequency to induce tetanus. Did the authors place the soleus muscle in a bath with a controlled temperature or not? This is because also the temperature is affecting picks fusion during tetanus. Considering this, the authors should perform again the experiment with a correct protocol (Delavar H, Nogueira L, Wagner PD, Hogan MC, Metzger D, Breen EC. Skeletal myofiber VEGF is essential for the exercise training response in adult mice. Am J Physiol Regul Integr Comp Physiol. 2014 Apr 15;306(8):R586-95. doi: 10.1152/ajpregu.00522.2013. Epub 2014 Feb 12. PMID: 24523345; PMCID: PMC4043130).

Regarding the figure, maximal values are lower compared to published literature (this can be due to the protocol used). In addition, why does the sample size between the absolute force and the normalized one change? Are these analyses been performed on different samples? They should be the same. Also, the authors did not see differences in muscle weight/body weight and fiber CSA. However, the differences between wt, R76W, and D130-136 mutant are exacerbated when absolute force is normalized on fiber area. Why this huge difference?

Response 4: Thanks for your comments and suggestions. In the present study, we used semi-isolated electrical stimulation model to determine skeletal muscle force. Mice were placed on a temperature-controlled platform to maintain normal body temperature during the complete experimental procedure. The distal tendon of soleus was surgically exposed, dissected, and attached to force transducer (MLT0420, Panlab, Austrilia) while the proximal end of soleus was preserved. The muscle was stimulated by applying electrical pulses through parallel platinum field electrodes placed on the proximal portion (motor point) of soles (Shaikh et al. 2020). To maintain physiological function of muscle, warm saline solution was continuously added on muscle surface during the muscle force measurement. Therefore, the electrodes were not directly touch in nerve or muscle fibers itself but stimulated the muscle through surrounding fluid. In addition, the condition that we used 25ms duration and 40ms interval for a stimulus was to obtain maximal muscle force from a multiple stimulus frequency ranges (Li et al. 2021). According to your suggestions, we should use aged transgenic mice at age of 18 months to perform again the experiment with the protocols from Breen’s lab, however, it is hard to realize now. On the one hand, the revised files are required to return within 10 days while it will take at least one year and a half to breed and raise aged transgenic mice. On the other hand, the labs of our school is closed these days due to the strict requirement to avoid the contamination of novel coronavirus (COVID-19) in China, including animal room. The open time is still not known, which depends on the progress of epidemic situation.

With regard to the figures, the sample size between the absolute force and the normalized one is the same. We are sorry for the mistake and correct them in the figure. Also, the specific isometric force or tetanic force was normalized to muscle cross-section area, not fiber area. We hope that the corrections will meet with approval.

[1] Shaikh, A., Fang, H., Li, M., Chen, S., Shang, P., Shang, X. Reduced expression of carbonic anhydrase III in skeletal muscles could be linked to muscle fatigue: A rat muscle fatigue model. Journal of Orthopaedic Translation, 2020, 22: 116-123.

[2] Li, G., Zhang, L., Ning, K., Yang, B., Acosta, F. M., Shang, P, et al., Osteocytic Connexin43 Channels Regulate Bone–Muscle Crosstalk. Cells. 2021.10(2), 237.

  1. Paragraph 2.6: The authors said that Cx43 channels impairment alters mitochondrial function. However, this is a hypothesis based on ROS content and expression of proteins involved in mitochondrial dynamics. They do not perform any assay for the analysis of mitochondrial functionality. The authors should measure membrane potential through TMRM and mitochondrial respiration. And they should include also adult mice as control.

In addition, the authors conclude that mitochondrial impairment and ROS content can be responsible for muscle weakness. But, ROS levels are higher also in R76W mutant where muscle force is not affected. Do the authors have an explanation for this? Moreover, would an anti-oxidant treatment be able to restore muscle force to normal levels in D130-136 mutant?

Response 5: Thanks for your comments. Evidence showed that aging process is related with increased reactive oxygen species (ROS) content and a reduction in endogenous antioxidant in the skeletal muscle (Reid et al. 2002). ROS have long been considered as deleterious species affecting skeletal muscle tissues. Indeed, increase in ROS can elicit oxidative modification of lipids, proteins and nucleic acids, leading to mitochondrial dysfunction and/or molecular damage (Elena et al. 2012). Thus, the accumulation of ROS and decline of antioxidants may directly, or indirectly, influence on mitochondrial function and consequently excitation-contraction coupling in aging muscle. It is a constructive suggestion to perform assay for the analysis of mitochondrial functionality, including membrane potential and mitochondrial respiration. These confirmatory study should be better investigated in the future.

In the study, ROS levels are higher in R76W mutant but muscle force is not affected. We think that the high ROS content affecting mitochondrial function may occur at the cellular level, and that these alterations have not yet translated into muscle force changes in aged R76W mice. In contrast, a decrease in muscle force in aged Δ130-136 mice may result from the combined effects of increased collagen deposition and high ROS level in aging muscle. Indeed, an anti-oxidant treatment in Δ130-136 mutant may be more helpful to verify the observation.  Thanks for your constructive suggestions.

[1] Reid, M., Durham, W. Generation of reactive oxygen and nitrogen species in contracting skeletal muscle: potential impact on aging. Ann. N. Y. Acad. Sci, 2002, 959, 108–116.

[2] Elena, Barbieri, Piero, Sestili. Reactive oxygen species in skeletal muscle signaling. J. Signal Transduct. 2012, 982794

  1. It is not clear to me why the authors decided to put the section on in vitro C2C12 differentiation with osteocyte CM, also because the effect that they saw on myotube diameter is not in line with CSA measurements. Considering that the 2 models and analysis cannot be compared and that this experiment does not add anything to the story, in my opinion, I would suggest removing it. In case, they decided to maintain it, I suggest putting more evident images of myotubes IF

Response 6: Thanks for your suggestions. We have removed the section on in vitro C2C12 differentiation with osteocyte CM from the text.

  1. In the abstract, the authors said that osteocyte apoptosis in aged mice is due to reduced PGE2 levels. This is speculation because they did not check if PGE2 supplementation leads to decreased osteocyte apoptosis since they performed the experiment only in C2C12 myotubes. Also, the authors concluded that the decreased PGE2 levels due to Cx43 impairment are responsible for muscle fibrosis in aged mice. But, did the authors check the levels of PGE2 in the circulation of the 3 mouse models in young and aged mice? Is it possible that PGE2 secreted from osteocytes is affecting other tissues, like muscle, in this case, considering that it is also produced by other organs? And if the authors find it to be decreased in the circulation, could a restoration of PGE2 to normal levels revert the fibrotic phenotype in aged muscles? And what about muscle weakness?

Response 7: Thanks for your comments. As you said, PGE2, an important intracellular messenger in the regulation of signal transduction, can be secreted from multiple organs or tissues, including muscle cells. However, PGE2 production by osteocytes is more than 100 times than PGE2 secretion by muscle cells (Bonewald, 2019). Also, gap junction and hemichannels are two major portals of PGE2 release in cell-to-cell communication and cell-to-extracellular matrix, respectively. To this end, we determined PGE2 content in primary osteocytes conditioned media (PO CM) collected from aged transgenic mice (Fig. 6f). Based on the results, we speculated that PGE2 may involve in bone and muscle phenotypes. According to our and other’s in vitro studies, PGE2 released by osteocytes mediated by Cx43 hemichannels protect against glucocorticoid-induced osteocyte apoptosis (Cherian, et al. 2005; Kitase, et al. 2010). In vivo, it is indeed a good idea to restore PGE2 to normal levels test whether it can alleviate the fibrotic phenotype. In this study, small sample size of animals is an important limiting factor affecting in vivo rescue experiments due to the low fertility rate and high mortality of transgenic mice during aging. If possible, we will further perform rescue experiment in vivo to test whether PGE2 supplementation revert muscle fibrotic phenotypes or muscle weakness in aged mice. Thanks again for your valuable suggestions to improve the quality of paper.  

[1] Bonewald, L. Use it or lose it to age: A review of bone and muscle communication. Bone, 2019, 120: 212-218.

[2] Cherian, P., Siller-Jackson, A, Gu S, et al. Mechanical strain opens connexin 43 hemichannels in osteocytes: a novel mechanism for the release of prostaglandin. Mol Biol Cell. 2005;16: 3100–6.

[3] Kitase, Y., Barragan, L., Qiang, H, et al. Mechanical induction of PGE2 in osteocytes blocks glucocorticoid-induced apoptosis through both the β-catenin and PKA pathways. J Bone Miner Res. 2010;25: 2381–92.

  1. PGE2 is also controlling muscle regeneration and MuSC function. Did the authors check for these parameters in their mice?

Response8: In this study, we did not examine muscle regeneration or MuSC function-related parameters. This is a subject to our continuing investigation in the future.

  1. a list of abbreviations should be added at the beginning of the manuscript or at least, the first time you write an abbreviation the authors should describe it. 

Response9: Thanks for your suggestion. We have provided a list of abbreviations in the revisions.

  1. in line with this, in Paragraph 2.3, the authors should explain what Alp is and why they decided to show NFAT-c, ATP and Alp in this context. 

Response10: Alp refers to alkaline phosphatase, a marker of bone formation. In this study, we analyzed the mRNA expression of ALP and NFAT-c in order to estimate bone formation and bone resorption in bone tissue. ATP is a regulatory factor for osteoclast formation and bone resorption. The full names of abbreviations (NFAT-c, ATP and Alp) have been provided in the text.

  1. line 458: pulsesnot pluses

Response11: We are sorry for the mistake. We have corrected the misspelling. 

  1. line 150: maybe it is osteoclast numberrather than osteoblastn. that is changing when GJs are inhibited

Response12: We have modified the sentence in the revision.

13.Trap staining for osteoclast should be more evident. Images are not clear

Response13: Thanks for your comment. We have placed the more evident images for Trap staining in the revision.

  1. Graphs color should be changed: in the blue bars the sample size is difficult to read

Response14: Thanks for your suggestion. We have changed the graphs color in the revision.

  1. Figure S3 should be ameliorated, maybe leaving all the muscle attached to the bone and cutting them together 

Response15: Thanks for your suggestion. Because of the difference in the way tissue sections were made between skeletal muscle and bone (as described in material and method 4.3), which may result in parts of the muscle tissue reserving incomplete in the picture. However, we think the Figure S3 could show the expression of GFP was predominantly detected in osteocytes but not surrounding muscle tissues.

Round 2

Reviewer 2 Report

Dear authors,

thank you for your exhaustive answers to my comments and your modifications to the manuscript accordingly. However, I suggest doing major experiments to address some of, at least for me, key points.

1.     I noticed that some asterisks have been added in Fig. 1 without changing the sample size. Do you have an explanation for this change?

2.     I also suggest putting the young controls, since you already have the data from previous works, in order to better appreciate the defects in bone and muscle mass that occurred during aging. 

3.     To me it is also important to perform at least one assay for mitochondrial functionality. Otherwise, the authors should rephrase the paragraph

Author Response

Dear editor and reviewer: 

We very much appreciate your valuable comments and suggestions to improve the quality of our paper. We have carefully addressed your comments point by point and marked the changes in red in the revised manuscript. Please see the attachment. 
